# Inter-Spacecraft Rapid Transfer Alignment Based on Attitude Plus Angular Rate Matching Using Q-Learning Kalman Filter

**DOI:** 10.3390/s25092774

**Published:** 2025-04-27

**Authors:** Kai Xiong, Peng Zhou, Xiangyu Huang

**Affiliations:** Science and Technology on Space Intelligent Control Laboratory, Beijing Institute of Control Engineering, Beijing 100190, China; zhoupeng_xmu@163.com (P.Z.); huangxyhit@sina.com (X.H.)

**Keywords:** spacecraft, attitude determination, transfer alignment, Q-learning, Kalman filter

## Abstract

This study focuses on the transfer alignment issue between a master spacecraft and a slave spacecraft for the scenario in which the slave spacecraft is mounted on the master satellite before release and should be ready to depart and perform its space mission independently. The challenge of the transfer alignment is to estimate the attitude and calibration parameters of the gyroscope unit (GU) on the slave spacecraft based on the attitude determination system (ADS) of the master spacecraft. To improve the accuracy and rapidity of the transfer alignment, a novel attitude plus angular rate matching scheme is presented using fused sensor information on the master spacecraft. Accordingly, a fifteen-dimensional state-space model is derived to estimate the spacecraft attitude, the GU bias, scale factor error and misalignment simultaneously. A Q-learning Kalman filter (QKF) is designed to fine tune the process noise covariance matrix related to the calibration parameters, which benefits the state estimation performance. The simulation results show that the presented attitude plus angular rate matching scheme performs better than the traditional attitude matching scheme, and the QKF outperforms the standard Kalman filter (KF) and the adaptive Kalman filter (AKF).

## 1. Introduction

Recently, there have been many space systems that contain the master spacecraft and the slave spacecraft, such as CubeSats deployed from the space station [1], the lander or rover released from a deep-space probe on the orbit of a planet [2] and the smart impactor launched from a mother flyby spacecraft [3]. Prior to the release of the slave spacecraft, its attitude has to be initialized and the systematic errors of its gyroscope unit (GU) should be calibrated, such that the slave spacecraft can perform its space mission independently. The attitude determination accuracy of the slave spacecraft depends on the calibration accuracy of the systematic errors, such as gyroscope bias, scale factor error and misalignment [4,5,6]. Although the ground calibration method is convenient to achieve high accuracy [7,8], the values of the calibration parameters may change due to the difference between the ground and space environments [9]. The primary cause for the change in the misalignment, which is the most dominant error that affects the attitude determination accuracy, is the thermal distortion of the GU bracket and the spacecraft body, which yields a pointing change in the GU axis in the space environment. To cope with this problem, the attitude determination system (ADS) of the master spacecraft can be utilized as the reference for the on-orbit calibration [10]. Typically, the high-precision gyroscopes and the star sensors are mounted on the master spacecraft for attitude determination [11,12,13]. An accurate and rapid transfer alignment is crucial to guarantee the attitude determination performance of the slave spacecraft [14,15].

The main problem of the traditional inter-spacecraft transfer alignment method is that the convergence rate of the algorithm is rather slow. Typically, tens of minutes are required for the traditional transfer alignment process [16], which is not satisfactory for quick response situations. To achieve faster convergence and increased accuracy, the rapid transfer alignment techniques are investigated in this paper.

The measurement information matching scheme is an important research object in the field of transfer alignment. Many scholars focus on the on-orbit calibration method for the GU based on the attitude matching scheme [17,18]. The traditional method is to estimate the attitude and the calibration parameters of the GU on the slave spacecraft using the attitude measurement data provided by the star sensors on the master spacecraft with a Kalman filter (KF) [19,20]. The high-accuracy measurement information obtained from the satellite payload and GNSS (global navigation satellite system) are taken into account for the calibration [21,22]. In addition, for tactical weapon applications, the velocity plus attitude matching scheme [23,24] is developed as an improvement of the conventional velocity matching scheme [25,26,27]. The calibration methods based on position matching and acceleration matching schemes are studied in [28,29].

On the basis of measurement information matching, the performance of the inter-spacecraft transfer alignment method depends on the filtering algorithm. The KF is the most widely used state estimation approach for the space missions [30,31,32]. The estimation accuracy of the KF depends on the process and measurement noise covariance matrices $Q_{k}$ and $R_{k}$ in the system model. If the prior statistical characteristics of the process and measurement noises are inaccurate, the performance of the KF may be degraded evidently. For the inter-spacecraft transfer alignment system, the statistical characteristics of the gyroscopes and the star sensors can be achieved from the manufacturer, while the process noise covariance related to the calibration parameters are often not known exactly. To cope with the problem, the common approach is to estimate the unknown noise covariance matrix with an adaptive Kalman filter (AKF) [33,34]. In the AKF, the estimate of $Q_{k}$ is updated recursively based on the measurement innovation, which is calculated with the previous state estimate in each iteration. However, when the previous state estimate is inaccurate, and the prior statistical information is far from the actual situation, it is difficult to obtain the proper $Q_{k}$ with the AKF [35]. To the best of the authors’ knowledge, the optimal approach to tune the process noise covariance matrix is still an open question.

Motivated by the key idea of the transfer alignment method for tactical weapon applications and the AKF algorithm, to improve the accuracy and rapidity of the inter-spacecraft transfer alignment, a practical method is presented in this paper. The main contributions of the paper are given as follows:(1)An attitude plus angular rate matching scheme is presented, where the fused information from the star sensors and the gyroscopes on the master spacecraft is adopted for the calibration of the GU on the slave spacecraft. Accordingly, the state equation and the measurement equation are derived as the transfer alignment system model. Compared with the traditional attitude matching scheme, the main advantage of the presented scheme is that more measurement information is utilized, such that the alignment performance is improved in limited time.(2)A framework of the Q-learning Kalman filter (QKF) that combines the celebrated Q-learning approach [36,37,38] with the KF is developed to fine tune the process noise covariance matrix related to the calibration parameters. Instead of the recursive estimation in the AKF, the Q-learning approach is designed to explore for the appropriate $Q_{k}$. Once the appropriate $Q_{k}$ is learned, it is plugged into the model-based KF to enhance its performance. Compared with our previous works [39,40], the main advantage of the presented QKF is that only one explorative filter (instead of a group of parallel filters) is required in the Q-learning process so as to simplify the implementation of the algorithm. Correspondingly, the computational load of the algorithm is decreased, which facilitates the application of the algorithm on the spacecraft with limited computing resource.

This study is structured as follows. In Section 2, the attitude plus angular rate matching scheme and the inter-spacecraft rapid transfer alignment system model are presented. In Section 3, the QKF algorithm for rapid transfer alignment is provided. In Section 4, the potential performance of the transfer alignment system is analyzed via the calculation of the Cramer–Rao lower bounds (CRLB) [41]. In Section 5, the estimation performance of the KF, the AKF and the presented QKF designed based on the transfer alignment system model are compared via simulations. Finally, the conclusions are drawn in the last section.

## 2. Attitude Plus Angular Rate Matching Scheme

### 2.1. Main Idea

The inter-spacecraft rapid transfer alignment model is derived to calibrate the GU on the slave spacecraft based on the ADS on the master spacecraft. The basic principle of the considered transfer alignment system is illustrated in Figure 1.

As shown in Figure 1, on the master spacecraft, the ADS consisting of gyroscopes and star sensors are utilized to obtain the attitude reference information. On the slave spacecraft, the transfer alignment filter incorporates the attitude reference information and the GU measurement to estimate the attitude, the GU bias and the calibration parameters. The transfer alignment filter is designed based on the transfer alignment model composed of the state equation and the measurement equation. The calibration parameters include the scale factor error and the misalignment of the GU on the slave spacecraft. The construction of the transfer alignment model and the parameterization of the systematic errors fit within the spacecraft attitude determination framework [10].

In this paper, the spacecraft attitude describes the rotation of the spacecraft body frame relative to the geocentric equatorial inertial frame. For the geocentric equatorial inertial frame, the origin is the Earth’s center, the X axis points to the Equinox direction, the Y axis points toward the North pole and the Z axis forms a right-handed coordinate with the X axis and Y axis. For the spacecraft body frame, the origin is the center mass of the spacecraft, and the X axis, Y axis and Z axis are parallel to the principal axis of inertia and form a right-handed coordinate.

### 2.2. State Equation

For the design of the transfer alignment filter, the state equation of the transfer alignment model is established based on the attitude kinematics and the error model of the GU on the slave spacecraft. In this paper, the GU is supposed to be three gyroscopes which measure the angular rate of the slave spacecraft relative to the inertial frame. The GU error model includes gyroscope bias, scale factor error, misalignment and random noise composed of angle random walk (ARW) and rate random walk (RRW), as shown in Figure 2.

In the GU error model, the measured angular rate $\omega_{gk}$ is related to the true angular rate $\omega_{bk}$ of the slave spacecraft relative to inertial frame in the body frame by [21](1)$$\omega_{gk}=\left(I+\Lambda_{k}\right)\left(I+\Delta_{k}\right)C_{b}^{g}\omega_{bk}+b_{k}+\eta_{ak},$$
where I denotes the unit matrix with compatible dimension, $\Lambda_{k}$ is the scale factor error matrix, $\Delta_{k}$ is the misalignment matrix, $C_{b}^{g}$ is the attitude transformation matrix from the body frame to the sensor frame, $b_{k}$ is the GU bias, $\eta_{ak}$ is the random noise called angular random walk, the subscript k denotes discrete time. The true angular rate is written as $\omega_{bk}={\left[\begin{array}{ccc} \omega_{bxk} & \omega_{byk} & \omega_{bzk} \end{array}\right]}^{T}$, where $\omega_{bxk}$, $\omega_{byk}$ and $\omega_{bzk}$ are the elements of 3-axis angular rate. The scale factor error matrix is given by(2)$$\Lambda_{k}=\left[\begin{array}{ccc} \lambda_{xk} & 0 & 0\\ 0 & \lambda_{yk} & 0\\ 0 & 0 & \lambda_{zk} \end{array}\right],$$
where $\lambda_{xk}$, $\lambda_{yk}$ and $\lambda_{zk}$ are the scale factor error parameters. The misalignment matrix is described as(3)$$\Delta_{k}=\left[\begin{array}{ccc} 0 & {\bar{\delta }}_{xyk} & {\bar{\delta }}_{xzk}\\ {\bar{\delta }}_{yxk} & 0 & {\bar{\delta }}_{yzk}\\ {\bar{\delta }}_{zxk} & {\bar{\delta }}_{zyk} & 0 \end{array}\right],$$
where ${\bar{\delta }}_{xyk}$, ${\bar{\delta }}_{xzk}$, ${\bar{\delta }}_{yxk}$, ${\bar{\delta }}_{yzk}$, ${\bar{\delta }}_{zxk}$ and ${\bar{\delta }}_{zyk}$ are elements in the misalignment matrix. From Equations (2) and (3), we have(4)$${(I+\Lambda }_{k})\left(I+\Delta_{k}\right)={I+M}_{k},$$
with(5)$$M_{k}=\left[\begin{array}{ccc} \lambda_{xk} & \delta_{xyk} & \delta_{xzk}\\ \delta_{yxk} & \lambda_{yk} & \delta_{yzk}\\ \delta_{zxk} & \delta_{zyk} & \lambda_{zk} \end{array}\right],$$
where(6)$$\delta_{xyk}=(1+\lambda_{xk}){\bar{\delta }}_{xyk},$$(7)$$\delta_{xzk}=(1+\lambda_{xk}){\bar{\delta }}_{xzk},$$(8)$$\delta_{yxk}=\left(1+\lambda_{yk}\right){\bar{\delta }}_{yxk},$$(9)$$\delta_{yzk}=(1+\lambda_{yk}){\bar{\delta }}_{yzk},$$(10)$$\delta_{zxk}=(1+\lambda_{zk}){\bar{\delta }}_{zxk},$$(11)$$\delta_{zyk}=\left(1+\lambda_{zk}\right){\bar{\delta }}_{zyk}.$$$\delta_{xyk}$, $\delta_{xzk}$, $\delta_{yxk}$, $\delta_{yzk}$, $\delta_{zxk}$ and $\delta_{zyk}$ are the misalignment parameters. Note that the products of scale factors and misalignments are combined in Equation (5) to simplify the formulation. Substituting Equation (4) into Equation (1), the GU error model is reformulated as(12)$$\omega_{gk}=\left({I+M}_{k}\right)C_{b}^{g}\omega_{bk}+b_{k}+\eta_{ak}.$$

To derive the attitude error equation, the estimate of the spacecraft angular rate ${\hat{\omega }}_{bk}$ is written as(13)$${\hat{\omega }}_{bk}=C_{g}^{b}(\omega_{gk}-{\hat{b}}_{k}),$$
where ${\hat{b}}_{k}$ is the estimate of the GU bias, $C_{g}^{b}$ is the attitude transformation matrix from the sensor frame to the body frame. Substituting Equation (12) into Equation (13), we have(14)$${\hat{\omega }}_{bk}=C_{g}^{b}\left({I+M}_{k}\right)C_{b}^{g}\omega_{bk}+C_{g}^{b}{\delta b}_{k}+C_{g}^{b}\eta_{ak},$$
where(15)$${\delta b}_{k}=b_{k}-{\hat{b}}_{k},$$
is the GU bias error. The angular rate estimation error is defined as(16)$${\delta \omega }_{bk}=\omega_{bk}-{\hat{\omega }}_{bk}.$$

Substituting Equation (14) into Equation (16) yields(17)$${\delta \omega }_{bk}=-C_{g}^{b}M_{k}C_{b}^{g}\omega_{bk}-C_{g}^{b}{\delta b}_{k}-C_{g}^{b}\eta_{ak}.$$

Equation (17) is reformulated as(18)$${\delta \omega }_{bk}=-C_{g}^{b}\Omega_{gk}\delta_{k}-C_{g}^{b}{\delta b}_{k}-C_{g}^{b}\eta_{ak},$$
with(19)$$\Omega_{gk}=\left[\begin{array}{ccc} {\left(C_{b}^{g}\omega_{bk}\right)}^{T} & 0_{1\times 3} & 0_{1\times 3}\\ 0_{1\times 3} & {\left(C_{b}^{g}\omega_{bk}\right)}^{T} & 0_{1\times 3}\\ 0_{1\times 3} & 0_{1\times 3} & {\left(C_{b}^{g}\omega_{bk}\right)}^{T} \end{array}\right]\;,$$
and(20)$$\delta_{k}={\left[\begin{array}{ccc} \begin{array}{ccc} \lambda_{xk} & \delta_{xyk} & \delta_{xzk} \end{array} & \begin{array}{ccc} \delta_{yxk} & \lambda_{yk} & \delta_{yzk} \end{array} & \begin{array}{ccc} \delta_{zxk} & \delta_{zyk} & \lambda_{zk} \end{array} \end{array}\right]}^{T}$$
is the calibration parameter vector.

For the spacecraft attitude determination, the error quaternion $\delta q_{k}$ is defined as(21)$$\delta q_{k}=q_{k}⨂{\hat{q}}_{k}^{-1},$$
where $q_{k}$ is the attitude quaternion that describes the spacecraft attitude relative to the inertial frame, ${\hat{q}}_{k}$ is the estimate of $q_{k}$. According to the spacecraft attitude kinematics, the propagation of the error quaternion is described by the following perturbation equation [11]:(22)$$\delta \rho_{k+1}=\left(I-\tau \left[\omega_{bk}\times \right]\right)\delta \rho_{k}+\frac{\tau }{2}{\delta \omega }_{bk},$$
where $\delta \rho_{k}$ is the vector part of the error quaternion, τ denotes the time interval of discretization, $\left[\omega_{bk}\times \right]$ is the skew symmetric matrix defined as(23)$$\left[\omega_{bk}\times \right]=\left[\begin{array}{ccc} 0 & -\omega_{bzk} & \omega_{byk}\\ \omega_{bzk} & 0 & -\omega_{bxk}\\ -\omega_{byk} & \omega_{bxk} & 0 \end{array}\right].$$

Substituting Equation (18) into Equation (22), the perturbation equation is modified as(24)$$\delta \rho_{k+1}=\left(I-\tau \left[\omega_{bk}\times \right]\right)\delta \rho_{k}-\frac{\tau }{2}C_{g}^{b}\Omega_{gk}\delta_{k}-\frac{\tau }{2}C_{g}^{b}{\delta b}_{k}-\frac{\tau }{2}C_{g}^{b}\eta_{ak}.$$

It is evident that the effects of the GU bias, scale factor error, misalignment and random noise to the propagation of the vector part of the error quaternion are described in Equation (24).

For transfer alignment, the spacecraft attitude, the GU bias, scale factor error and misalignment should be estimated. Accordingly, the state vector is constructed as the combination of the vector part of the error quaternion $\delta \rho_{k}$, the GU bias error ${\delta b}_{k}$ and the calibration parameter vector $\delta_{k}$, which is given by(25)$$x_{k}={\left[\begin{array}{ccc} {\delta \rho }_{k}^{T} & \delta b_{k}^{T} & \delta_{k}^{T} \end{array}\right]}^{T}.$$

From Equations (24) and (25), we obtain the state equation:(26)$$x_{k+1}=F_{k}x_{k}+w_{k},$$
with the state transition matrix(27)$$F_{k}=\left[\begin{array}{ccc} I_{3\times 3}-\tau \left[\omega_{bk}\times \right] & -\frac{\tau }{2}C_{g}^{b} & -\frac{\tau }{2}C_{g}^{b}\Omega_{gk}\\ 0_{3\times 3} & I_{3\times 3} & 0_{3\times 9}\\ 0_{9\times 3} & 0_{9\times 3} & I_{9\times 9} \end{array}\right].$$

The process noise $w_{k}$ is given by(28)$$w_{k}={\left[\begin{array}{ccc} -\frac{\tau }{2}{\left(C_{g}^{b}\eta_{ak}\right)}^{T} & \eta_{rk}^{T} & \eta_{ck}^{T} \end{array}\right]}^{T},$$
where $\eta_{rk}$ is the random noise that drive the rate random walk, $\eta_{ck}$ is introduced to describe the drift of the calibration parameters. It is often assumed that $w_{k}$ is the Gaussian white noise with zero mean. The process noise covariance matrix $Q_{k}$ is a positive definite symmetric matrix with the following structure(29)$$Q_{k}=E\left(\left[\begin{array}{c} -\frac{\tau }{2}C_{g}^{b}\eta_{ak}\\ \eta_{rk}\\ \eta_{ck} \end{array}\right]\left[\begin{array}{ccc} -\frac{\tau }{2}C_{g}^{b}\eta_{ak} & \eta_{rk} & \eta_{ck} \end{array}\right]\right)=\left[\begin{array}{ccc} Q_{ak} & 0 & 0\\ 0 & Q_{rk} & 0\\ 0 & 0 & Q_{ck} \end{array}\right],$$
where $Q_{ak}$ and $Q_{rk}$ are the sub-matrices related to the vector part of the error quaternion $\delta \rho_{k}$ and the GU bias error ${\delta b}_{k}$ respectively, $Q_{ck}$ is the sub-matrix related to the calibration parameter vector $\delta_{k}$.

### 2.3. Measurement Model

For the attitude plus angular rate matching scheme, both the attitude and the angular rate reference information achieved from the ADS on the master spacecraft are utilized for the on-orbit calibration. It is expected that the calibration parameters can be estimated effectively with the attitude reference information from the master spacecraft. When the attitude and the angular reference information is available, the measurement equation is written as(30)$$y_{k}=H_{k}x_{k}+v_{k},$$
with the measurement(31)$$y_{k}={\left[\begin{array}{cc} y_{\rho k}^{T} & y_{\omega k}^{T} \end{array}\right]}^{T},$$
the measurement matrix(32)$$H_{k}=\left[\begin{array}{ccc} I_{3\times 3} & 0_{3\times 3} & 0_{3\times 9}\\ 0_{3\times 3} & -C_{g}^{b} & -C_{g}^{b}\Omega_{gk} \end{array}\right],$$
and the measurement noise(33)$$v_{k}={\left[\begin{array}{cc} v_{\rho k}^{T} & v_{\omega k}^{T} \end{array}\right]}^{T},$$
where $y_{\rho k}$ is the vector part of the error quaternion between the quaternions obtained from the ADS on the master spacecraft and the GU on the slave spacecraft, $y_{\omega k}$ is the difference between the angular rates obtained from the ADS on the master spacecraft and the GU on the slave spacecraft, $v_{\rho k}$ is the attitude measurement noise with the covariance matrix $R_{\rho k}$, $v_{\omega k}$ is the angular rate measurement noise with the covariance matrix $R_{\omega k}$. The measurement noise covariance matrix $R_{k}$ is the mixture of $R_{\rho k}$ and $R_{\omega k}$:(34)$$R_{k}=\left[\begin{array}{cc} R_{\rho k} & 0\\ 0 & R_{\omega k} \end{array}\right].$$

Note that both the measurement noises of the ADS on the master spacecraft and the GU on the slave spacecraft are contained in Equation (33). Generally, to implement the inter-spacecraft rapid transfer alignment effectively, the ADS on the master spacecraft should be more accurate than the GU on the slave spacecraft. It is expected that the systematic errors in the ADS on the master spacecraft have been compensated before the implementation of the inter-spacecraft rapid transfer alignment.

The state Equation (26) and the measurement Equation (30) compose the system model for the attitude plus angular rate matching scheme. With the system model, the transfer alignment KF is designed to implement the state estimation. It should be mentioned that, for the considered attitude determination system composed of the star sensors and the gyroscopes, as it is widely used in current satellites, the feasibility of the system model has been verified through multiple space missions. In general, for a novel navigation system, the hardware-in-loop experiment with the simulation of the operational environment is an effective approach for model verification.

### 2.4. Transfer Alignment KF

On the basis of the system model, the transfer alignment KF is designed to estimate the state vector $x_{k}$ based on the measurement $y_{k}$. The procedure of the inter-spacecraft rapid transfer alignment method based on the KF with the prediction and update procedures is collected in Algorithm 1.
**Algorithm 1:** Transfer alignment Kalman filter1: Initialize attitude quaternion estimate ${\hat{q}}_{0}$, bias estimate ${\hat{b}}_{0}$, state estimate ${\hat{x}}_{0}$ and its estimation error covariance matrix $P_{0}$
2: **for** *k* = 1, 2, …, K, **do**3: ${\bar{\omega }}_{bk-1}\leftarrow C_{g}^{b}{\left({I+\hat{M}}_{k-1}\right)}^{-1}(\omega_{gk-1}-{\hat{b}}_{k-1})$
4: ${\hat{q}}_{k}\leftarrow [I_{4\times 4}\cos\left(\frac{\left\|{\bar{\omega }}_{bk-1}\right\|\tau }{2}\right)+\frac{\Phi \left({\bar{\omega }}_{bk-1}\right)}{\left\|{\bar{\omega }}_{bk-1}\right\|}sin(\frac{\left\|{\bar{\omega }}_{bk-1}\right\|\tau }{2})]{\hat{q}}_{k-1}$
5: ${\hat{b}}_{k}\leftarrow {\hat{b}}_{k-1}$
6: ${\hat{x}}_{k|k-1}(1:6,1)\leftarrow 0$
7: ${\hat{x}}_{k|k-1}(7:15,1)\leftarrow {\hat{x}}_{k-1}(7:15,1)$
8: $P_{k|k-1}\leftarrow F_{k}P_{k-1}F_{k}^{T}+Q_{k}$
9: **if** the measurement $y_{k}$ is available, **then**
10:  $K_{k}\leftarrow P_{k|k-1}H_{k}^{T}(H_{k}P_{k|k-1}H_{k}^{T}{+R_{k})}^{-1}$
11:  ${\tilde{y}}_{k}\leftarrow y_{k}-H_{k}{\hat{x}}_{k|k-1}$
12:  ${\hat{x}}_{k}\leftarrow {\hat{x}}_{k|k-1}+K_{k}{\tilde{y}}_{k}$
13:  $P_{k}\leftarrow \left(I-K_{k}H_{k}\right)P_{k|k-1}{\left(I-K_{k}H_{k}\right)}^{T}+K_{k}R_{k}K_{k}^{T}$
14: **end if**15: ${\hat{q}}_{k}\leftarrow {\delta \hat{q}}_{k}⨂{\hat{q}}_{k}$
16: ${\hat{b}}_{k}\leftarrow {\hat{b}}_{k}+{\delta \hat{b}}_{k}$
17: **end for**18: **return** $\left\{{\hat{q}}_{k}\right\}$, $\left\{{\hat{b}}_{k}\right\}$, $\left\{{\hat{x}}_{k}\right\}$, $\left\{P_{k}\right\}$


In the algorithm, ${\hat{x}}_{k|k-1}$ and ${\hat{x}}_{k}$ are the prediction and the estimate of the state vector, $P_{k|k-1}$ and $P_{k}$ are their corresponding estimation error covariance matrices, K is the length of the measurement data, $K_{k}$ is the Kalman gain, ${\tilde{y}}_{k}$ is the measurement innovation. The matrix ${\hat{M}}_{k}$ is constructed with the estimate of the calibration parameter vector as the estimate of $M_{k}$. The expression of $\Phi \left({\bar{\omega }}_{bk}\right)$ is(35)$$\Phi \left({\bar{\omega }}_{bk}\right)=\left[\begin{array}{cc} -[{\bar{\omega }}_{bk}\times ] & {\bar{\omega }}_{bk}\\ -{\bar{\omega }}_{bk}^{T} & 0 \end{array}\right].$$

To facilitate the implementation of the algorithm, similar to the method presented in [11], as the attitude quaternion ${\hat{q}}_{k}$ is propagated based on the spacecraft attitude kinematics equation, the state transition matrix $F_{k}$ in the state Equation (26) is only used for the estimation error covariance propagation in the KF. Note that the propagation of the error quaternion ${\delta \hat{q}}_{k}$ is not beneficial to improve the filtering performance. To deal with the problem caused by the discontinuity of the measurement data, for the filtering algorithm shown in Algorithm 1, the propagation is performed in each time step, while the update is performed in the case that the measurement data are available.

It is seen from Algorithm 1 that the efficiency of the update to the prediction ${\hat{x}}_{k|k-1}$ with the measurement innovation ${\tilde{y}}_{k}$ depends on the Kalman gain $K_{k}$, which is adjusted through the noise covariance matrices $Q_{k}$ and $R_{k}$. In the measurement noise covariance matrix, $R_{\rho k}$ and $R_{\omega k}$ are symmetric positive definite matrices determined according to the measurement error behavior of the star sensors and the GU specified by the manufacturer. In the process noise covariance matrix, the elements related to $\eta_{ak}$ and $\eta_{rk}$ are determined with the ARW and RRW coefficients, which are achievable through the Allan variance analysis [42]. However, it is difficult to determine the accurate noise covariance for the calibration parameters $\eta_{ck}$ in the absence of prior knowledge. As mentioned in the introduction, the proper choice for the process noise covariance matrix is critical for accurate transfer alignment. The state estimate of the KF may deviate from its actual value if $Q_{k}$ is not set appropriately. In order to identify the unknown elements in $Q_{k}$, a Q-learning-based filtering algorithm is presented in the next section as a modification of the standard KF.

## 3. Q-Learning Kalman Filter

### 3.1. Q-Learning Approach

When aiming to solve the problem of fine tuning the elements related to the calibration parameters in the process noise covariance matrix of the transfer alignment KF, a Q-learning Kalman filter is presented, which is a combination of the KF algorithm and the Q-learning approach. The Q-learning approach is a representative reinforce learning (RL) method [43], which is constructed on the key idea that the successful decision should be remembered by the agent interacting with its environment, which could provide a reinforcement signal as the feedback to the successful decision of the agent, such that the decision becomes more likely to be made in the future.

The Q-learning approach is implemented through a recursive trial-and-error process. In each iteration, the agent performs an action and receives an immediate reward from the environment, which indicates whether the action is good or not. The reward is accumulated to update the action selection strategy represented with the Q-function. Then, an optimized action is selected according to the updated strategy for the next iteration. This process is repeated several times so that the appropriate action for the maximization of the accumulated reward tends to be selected by the agent. The model of the environment is not necessary in the Q-learning process, which facilitates the implementation of the approach. The Q-learning approach has become the basis of many learning algorithms and exhibits an excellent learning ability.

The purpose of the Q-learning approach is to achieve the proper strategy to select the action a∈A in a specific state s∈S, where A and S are the action space and the state space, respectively. To achieve the purpose, the iterative update of the Q-function $Q^{\left(k\right)}(s,a)$ is formulated as(36)$$Q^{\left(k\right)}\left(s,a\right)=\left(1-\alpha \right)Q^{\left(k-1\right)}\left(s,a\right)+\alpha \left[R\left(s,a\right)+\gamma \underset{a^{’}\in A}{\max}Q^{\left(k-1\right)}\left(s^{\prime },a^{\prime }\right)\right],$$
where $R\left(s,a\right)$ is the immediate reward for the action a that is performed in the state s, 0<α≤1 is the learning rate, 0<γ≤1 is the discount factor and s’ is the transited state after the action a is performed. It is seen from (36) that the Q-learning process is the incremental estimation of the Q-function $Q^{\left(k\right)}(s,a)$. In the learning process, the discount factor γ is used to weigh between the immediate reward $R\left(s,a\right)$ and the accumulated reward represented by the previous estimation of the Q-function. The effect of the accumulated reward to the estimation of Q-function will be enhanced with the increase in the parameter γ. The learning rate α is used to weigh between the previous estimation of the Q-function $Q^{\left(k-1\right)}(s,a)$ and its update, as shown in (36). The effect of the update will be enhanced with the increase of the parameter α.

To guarantee the efficiency of the Q-learning approach, it is important to design the action space A, the state space S and the reward $R\left(s,a\right)$ properly. According to the Q-function $Q^{\left(k\right)}(s,a)$, the selected action $a_{max}$ for the next iteration can be determined as(37)$$a_{max}\leftarrow \arg\underset{a\in A}{\max}Q^{\left(k\right)}(s,a).$$

As the Q-function represents the accumulated reward, the action selection strategy shown in Equation (37) is beneficial to maximize the accumulated reward.

### 3.2. Q-Learning Based Covariance Tuning

The model-based KF and the data-driven Q-learning approach are combined to design the QKF algorithm, where the Q-learning approach is used instead of the recursive estimation in the AKF algorithm to determine the process noise covariance matrix. To simplify the Q-learning process, only the sub-matrix $Q_{ck}$ is tuned in the presented algorithm. The suggested framework of the QKF is shown in Figure 3.

The Q-learning approach is implemented based on the measurement data obtained from the ADS on the master spacecraft and the GU on the slave spacecraft. Once the sub-matrix corresponding to the calibration parameter vector is selected as ${\hat{Q}}_{ck}$ via the Q-learning approach, it is plugged into the transfer alignment KF presented in Algorithm 1 for the state estimation. The process noise covariance matrix of the transfer alignment KF is formulated as(38)$${\hat{Q}}_{k}=\left[\begin{array}{ccc} Q_{ak} & 0 & 0\\ 0 & Q_{rk} & 0\\ 0 & 0 & {\hat{Q}}_{ck} \end{array}\right].$$

It differs from that of the standard KF as the fine-tuned ${\hat{Q}}_{k}$ is adopted instead of the original $Q_{k}$ to calculate the gain matrix $K_{k}$. In other application scenarios, all sub-matrices in $Q_{k}$ should be tuned, a parallel Q-learning based filtering algorithm similar to the method developed in [40] could be adopted.

To select the appropriate process noise covariance matrix for filtering performance enhancement, in the designed Q-learning approach, the state space S is constructed with different design values of the sub-matrix $Q_{ck}^{\left(s\right)}$. Each state s is related to an element in the pre-determined set $\left\{\cdots ,Q_{ck}^{\left(s\right)},\cdots \right\}$. The action space A is constructed with different state transition actions, including the transition to the adjacent state or stay at the current state. Furthermore, the immediate reward $R\left(s,a\right)$ is constructed with the measurement innovation of an explorative KF designed based on the system model shown in (26) and (30). The structure and parameters of the explorative KF is similar to those of the transfer alignment KF, except the process noise covariance matrix related to the current state ${\bar{Q}}_{k}^{\left(s\right)}$ is adopted. Specifically, the process noise covariance matrix in the explorative KF is formulated as(39)$${\bar{Q}}_{k}^{\left(s\right)}=\left[\begin{array}{ccc} Q_{ak} & 0 & 0\\ 0 & Q_{rk} & 0\\ 0 & 0 & Q_{ck}^{\left(s\right)} \end{array}\right].$$

In the Q-learning process, for the state that transits from s to s’ after the action a is performed, the reward is designed as(40)$$R\left(s,a\right)={\left({\tilde{y}}_{e}^{\left(s\right)}\right)}^{T}{\tilde{y}}_{e}^{\left(s\right)}-{\left({\tilde{y}}_{e}^{\left(s’\right)}\right)}^{T}{\tilde{y}}_{e}^{\left(s’\right)},$$
where ${\tilde{y}}_{e}^{\left(s\right)}$ is the measurement innovation of the explorative KF with ${\bar{Q}}_{k}^{\left(s\right)}$ used instead of $Q_{k}$. From the reward represented in Equation (40), it is evident that a positive reward is achieved if the action a is beneficial to reduce the measurement innovation. Conversely, the action that increases the measurement innovation leads to negative feedback for the agent. Considering that measurement innovation is an indicator of filtering performance, it is expected that the sub-matrix ${\hat{Q}}_{ck},$ selected with the Q-learning approach via the maximization of the cumulative reward represented by the Q-function $Q\left(s,a\right),$ is valuable to improve the state estimation accuracy. Note that the subscript $\left(k\right)$ of the Q-function $Q^{\left(k\right)}(s,a)$ is omitted hereafter to simplify the notation.

Following the previous description, the Q-learning process in the QKF algorithm to achieve ${\hat{Q}}_{ck}$ is presented in Algorithm 2. To balance the exploration and exploitation in the Q-learning process, the ε–greedy strategy [43] is adopted as a modification of the basic action selection strategy shown in Equation (37).
**Algorithm 2**: Q-learning process in QKF1: ${\hat{x}}_{e0}\leftarrow {\hat{x}}_{0}$, $P_{e0}\leftarrow P_{0}$
2: Initialize parameter set $\left\{\cdots ,Q_{ck}^{\left(s\right)},\cdots \right\}$
3: **for** all s∈S, ${\bar{Q}}_{k}^{\left(s\right)}(1:6,1:6)\leftarrow Q_{k}(1:6,1:6)$, ${\bar{Q}}_{k}^{\left(s\right)}(7:15,7:15)\leftarrow Q_{ck}^{\left(s\right)}$
4: **for** all s∈S, a∈A, Q(s,a)←0
5: Initialize state s
6: ${\tilde{y}}_{e}^{\left(s\right)}\leftarrow sqrt(eig\left(H_{0}P_{e0}H_{0}^{T}+R_{0}\right))$
7: **for** *k* = 1, 2, …, K, **do**8:  a←ε–greedy(s,Q(s,a),A,ε)
9:  Perform a and observe reached state s’
10:  $\left[{\hat{x}}_{ek},P_{ek},{\tilde{y}}_{e}^{\left(s’\right)}]\leftarrow KF({\hat{x}}_{ek-1},P_{ek-1},y_{k},{\bar{Q}}_{k}^{\left(s’\right)},R_{k}\right)$
11:  $R\left(s,a\right)\leftarrow {\left({\tilde{y}}_{e}^{\left(s\right)}\right)}^{T}{\tilde{y}}_{e}^{\left(s\right)}-{\left({\tilde{y}}_{e}^{\left(s’\right)}\right)}^{T}{\tilde{y}}_{e}^{\left(s’\right)}$
12:  $Q\left(s,a\right)=\left(1-\alpha \right)Q\left(s,a\right)+\alpha [R\left(s,a\right)+\gamma \underset{\mathrm{a’}\in A}{\max}Q\left(s’,a’\right)]$
13:  s←s’
14:  ${\tilde{y}}_{e}^{\left(s\right)}\leftarrow {\tilde{y}}_{e}^{\left(s’\right)}$
15: **end for**16: ${\hat{Q}}_{ck}\leftarrow Q_{ck}^{\left(s\right)}$
17: **return** ${\hat{Q}}_{ck}$


In the algorithm, ${\hat{x}}_{ek}$ and $P_{ek}$ are the state estimate and its estimation error covariance matrix of the explorative KF, and ε is the probability of the ε-greedy strategy to select a random action. The function eig() denotes the eigenvalues of a square matrix. The function sqrt() denotes the square root of each element in a vector. The function ε–greedy() denotes the ε-greedy strategy, where the agent selects the random action in the action space A with the probability ε and selects the action that maximizes the Q-function, as shown in Equation (37) with the probability (1−ε). The function KF() is the KF equations, which are similar to the predication and update equations in Algorithm 1. The output ${\hat{Q}}_{ck}$ of the algorithm is exploited in the transfer alignment KF.

Generally, the computational load of the QKF is related to the number of the parallel KFs in the algorithm. From previous works, it is known that only a few explorative KFs are sufficient to improve the filtering performance evidently. In this paper, from Figure 3 and Algorithm 2, only one explorative KF and one transfer alignment KF are contained in the QKF algorithm. Thus, the computational load of the QKF is about two times larger than the standard KF. It is easy to complete the computation of the QKF in the interval of the measurement update. The moderate increase in the computational load is affordable for the current onboard computers.

To ensure the efficiency of the QKF algorithm, an important aspect is to design the bound of the state space S appropriately. It is expected that a dynamic state space with the bound stretched automatically can be designed in future works. On the basis of the QKF presented in Algorithm 2, the measurement innovation sequence in a time window could be taken into account to suppress the unfavorable effect of the measurement noise [40]. Although the process noise covariance matrix obtained from the algorithm may not be globally optimal, it is often an effective and simple approach to improve the filtering performance.

## 4. CRLB of Transfer Alignment System

The feasibility of the inter-spacecraft rapid transfer alignment method based on the attitude plus angular rate matching scheme is analyzed through the CRLB. The CRLB is a theoretical bound on the achievable state estimation accuracy for certain system models. It facilitates the potential performance analysis of the transfer alignment method before the numerical simulation of the filtering algorithm. For the linear discrete-time stochastic system formulated in Equations (26) and (30), the calculation process of the CRLB is described in Algorithm 3.
**Algorithm 3**: Calculation of CRLB1: $J_{0}\leftarrow P_{0}^{-1}$
2: **for** *k* = 1, 2, …, K, **do**3: $J_{k}\leftarrow {\left(F_{k}J_{k-1}^{-1}F_{k}^{T}+Q_{k}\right)}^{-1}$
4: **if** the measurement $y_{k}$ is available, **then**
5:  $J_{k}\leftarrow J_{k}+H_{k}^{T}R_{k}^{-1}H_{k}$
6: **end if**7: ${\hat{P}}_{k}\leftarrow J_{k}^{-1}$
8: **end for**
9: **return**
$\left\{{\hat{P}}_{k}\right\}$


In the algorithm, $J_{k}$ is the fisher information matrix calculated with the considered system model. The square roots of the diagonal elements of the calculated matrix ${\hat{P}}_{k}$ provide the theoretical bound of the state estimation root mean square (RMS) error.

The CRLB analysis is implemented under the following conditions. The master spacecraft is an Earth satellite with an orbit altitude of 700 km and its attitude keeps in orientation to the Earth. The slave spacecraft is mounted on the master spacecraft. For the ADS on the master spacecraft, the attitude determination accuracy is 3″ and the angular rate determination accuracy is 0.02°/h. For the GU on the slave spacecraft, the ARW and RRW coefficients are 4 × 10^−4^°/h^0.5^ and 1 × 10^−3^°/h^1.5^ respectively. The scale factor error parameter vector is set as ${\left[500 500 500\right]}^{T}ppm$ (parts per million) and the misalignment parameter vector is set as ${\left[\begin{array}{cc} \begin{array}{ccc} 50\prime \prime & 50\prime \prime & 50\prime \prime \end{array} & \begin{array}{ccc} 50\prime \prime & 50\prime \prime & 50\prime \prime \end{array} \end{array}\right]}^{T}$. The update rate of the filter is 1 Hz. The total simulation time of the transfer alignment is 3600 s. The calibration maneuver is the sequential rotation around the three orthogonal axes of the master spacecraft body frame. Generally, the transfer alignment performance can be improved when the calibration maneuver angular rate is increased. Considering that the dynamic measurement performance of the typical star sensor may be degraded if its angular rate is larger than 2°/s, the rotation angular rate of the master spacecraft is set as 1°/s.

The CRLB for the inter-spacecraft rapid transfer alignment system described in Section 2 is calculated using Algorithm 3. Figure 4, Figure 5 and Figure 6 give the theoretical error bounds of the calibration parameters when the time length of the rotation around each axis is 100 s, 200 s, 400 s and 600 s, respectively.

From the CRLB curves, it is evident that the rapidity of the transfer alignment is guaranteed when a shorter rotation time is adopted. According to the analysis results, in the following numerical simulation, the time length of the rotation around each axis is set as 100 s for the calibration maneuver.

## 5. Simulation Results

To illustrate the high performance of the inter-spacecraft rapid transfer alignment method based on the QKF, the simulation results are shown in this section. For the data generation of the sensors, the simulation conditions are same as those in Section 4. In the transfer alignment KF, the initial state estimation error covariance matrix is set as follows.(41)$$P_{0}=\left[\begin{array}{ccc} p_{a}^{2}I_{3\times 3} & & \\ & p_{r}^{2}I_{3\times 3} & \\ & & p_{c}^{2}I_{3\times 3} \end{array}\right],$$
where $p_{a}=0.05{}^{\circ}$, $p_{r}=0.05{}^{\circ}/h$. The elements in $p_{c}$ are larger than triple the magnitude of the calibration parameters given in Section 4. Similarly to the attitude determination KF [21], in the process noise covariance matrix $Q_{k}$, the sub-matrices $Q_{ak}$ and $Q_{rk}$ are designed according to the ARW and RRW coefficients of the GU. The sub-matrix $Q_{ck}$ is set as $Q_{ck}=q_{c}^{2}I_{9\times 9}$, where the magnitude of the parameter $q_{c}$ is $10^{-5}$. In the measurement noise covariance matrix $R_{k}$, the sub-matrices are set as $R_{\rho k}=r_{\rho }^{2}I_{3\times 3}$ and $R_{\omega k}=r_{\omega }^{2}I_{3\times 3}$, where $r_{\rho }=3\prime \prime$ and $r_{\omega }=0.02{}^{\circ}/h$. For the inter-spacecraft rapid transfer alignment, the initial attitude estimate ${\hat{q}}_{0}$ is obtained from the ADS on the master spacecraft, and the state vector related to the calibration parameters is initialized as zero.

The first simulation is performed to illustrate the high performance of the attitude plus angular rate matching scheme presented in Section 2. The presented attitude plus angular rate matching scheme is compared with the traditional attitude matching scheme via the simulation. The average RMS errors of the attitude plus angular rate matching scheme and the attitude matching scheme for the estimation of the attitude and the calibration parameters obtained from 10 individual trials are plotted together in Figure 7, Figure 8 and Figure 9.

It is seen from Figure 7, Figure 8 and Figure 9 that the presented scheme performs better than the traditional scheme. The reason is that more measurement information is available for the transfer alignment in the attitude plus angular rate matching scheme. With the simulation results shown in the figures above, we conclude that the presented attitude plus angular rate matching scheme is efficient for the inter-spacecraft transfer alignment.

The aim of the second simulation is to illustrate the performance of the QKF presented in Section 3. In the QKF algorithm, the Q-learning parameters are set as α=0.2, γ=0.8 and ε=0.5. Generally, the parameters can be designed with a trial-and-error method via the numerical simulation. For the considered noise covariance adaptation problem in the filtering algorithm, it was found in previous works [39,44] that the influence of the design parameters is not significant when they are chosen in certain scopes. In the pre-determined set related to the state space S for the tuning of the process noise covariance matrix, the sub-matrix $Q_{ck}^{\left(s\right)}$ is set as(42)$$Q_{ck}^{\left(s\right)}={\left(\lambda^{\left(s\right)}\right)}^{2}q_{c}^{2}I_{9\times 9},$$
where $\lambda^{\left(s\right)}$ is a scalar factor in the range of $\lambda^{\left(s\right)}\in [1,\;100]$, with 11 different values inside this interval. The cardinality of the pre-determined set is rather small, which is beneficial for the Q-learning approach in order to achieve a reasonable result. In fact, for the considered scenario, a small parameter set is effective to improve the filtering performance. Nevertheless, a sophisticated Q-learning approach may be required for more complicated problems.

The estimation error curves of the attitude and the calibration parameters as well as their corresponding ±3σ error bounds computed from the filter’s error covariance matrix are shown in Figure 10, Figure 11 and Figure 12.

It is seen from the figures that the estimation errors of the calibration parameters diminish rapidly after the calibration maneuver is performed. Both the scale factor and the misalignment estimation errors converge in about 600 s. All the estimation error curves of the QKF are contained in the corresponding error bounds, which indicate the consistency of the filtering algorithm. The simulation result illustrates that all the systematic errors of the GU on the slave spacecraft in the system model can be calibrated rapidly and accurately with the ADS on the master spacecraft. Simultaneously, it shows that the QKF is feasible for the inter-spacecraft rapid transfer alignment.

To facilitate the comparison, the average RMS errors of different transfer alignment methods are listed in Table 1.

Obviously, the presented attitude plus angular rate matching scheme outperforms the traditional attitude matching scheme. The state estimation accuracy can be improved when the presented QKF is used instead of the standard KF.

Furthermore, to demonstrate the advantage of the QKF algorithm, the simulation of the inter-spacecraft rapid transfer alignment methods based on different filtering algorithms are performed, including the standard KF, the Sage-Husa AKF and the QKF. For a fair comparison, three filtering algorithms share the basic filtering parameters, including the initialization parameters, the state transition matrix, the measurement matrix and the initial noise covariance matrices. The average RMS errors of the KF, the AKF and the QKF for the estimation of the attitude and the calibration parameters obtained from 10 individual trials are plotted in Figure 13, Figure 14 and Figure 15.

The above figures illustrate that the QKF obtains the highest state estimation accuracy in comparison with the KF and the AKF. This result is distinguishable in terms of both the attitude and the calibration parameters estimation. It indicates that the QKF is more effective than the AKF to identify the appropriate process noise covariance matrix and consequently improves the filtering performance.

To illustrate the performance of the presented QKF algorithm in different scenarios, the numerical simulation is implemented with different measurement noise levels. In the inter-spacecraft rapid transfer alignment system, when the attitude measurement noise standard deviation increases from 1″ to 10″, the average attitude estimation RMS errors of the KF and the QKF obtained from 10 individual trials are plotted in Figure 16. It indicates that the QKF algorithm is less sensitive to the variation in the measurement noise levels.

To sum up, it is apparent from the simulation results that the presented inter-spacecraft rapid transfer alignment method is valuable to satisfy the rapidity and accuracy requirements for the on-orbit calibration. The presented attitude plus angular rate matching scheme performs better than the traditional one. The QKF has considerable potential for practical applications in space missions as both the KF algorithm and the Q-learning approach are familiar for aerospace engineers. The basic principle of the presented method can be further applied to calibrate other attitude sensors, although the system model should be modified separately depending on the specific application.

## 6. Conclusions

The accurate and rapid transfer alignment is critical for the GU on the slave spacecraft, which influences the performance of the slave spacecraft after it is released from the master spacecraft. To satisfy the requirements of the space missions, this paper presents the novel inter-spacecraft rapid transfer alignment method based on the attitude plus angular rate matching scheme. For fine tuning the process noise covariance matrix of the transfer alignment KF, the Q-learning approach is incorporated with the model-based KF, which results in the QKF algorithm. Simulations are implemented for the performance evaluation. The simulation results demonstrate that the presented method can estimate all the calibration parameters in the transfer alignment model, and the misalignment estimation accuracy of less than 2 mrad is achievable within 10 min. The attitude plus angular rate matching scheme presents improved performance compared with the traditional attitude matching scheme. The QKF achieves the best estimation accuracy compared with the KF and AKF algorithms. Further work is required to evaluate the efficiency of the presented inter-spacecraft rapid transfer alignment method in practical applications and optimize the algorithm design according to the real error behavior. Although demonstrated for space missions, the presented method can be elaborated for information fusion and error calibration in other platforms for future research.

## Figures and Tables

**Figure 1 sensors-25-02774-f001:**
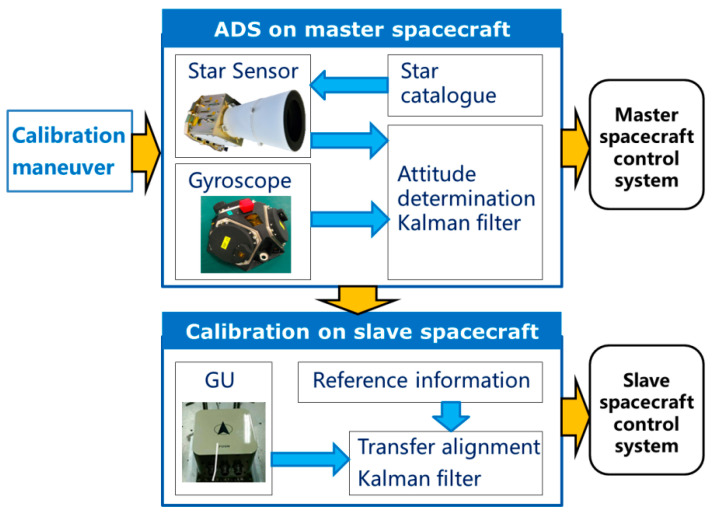
Diagram of inter-spacecraft rapid transfer alignment.

**Figure 2 sensors-25-02774-f002:**
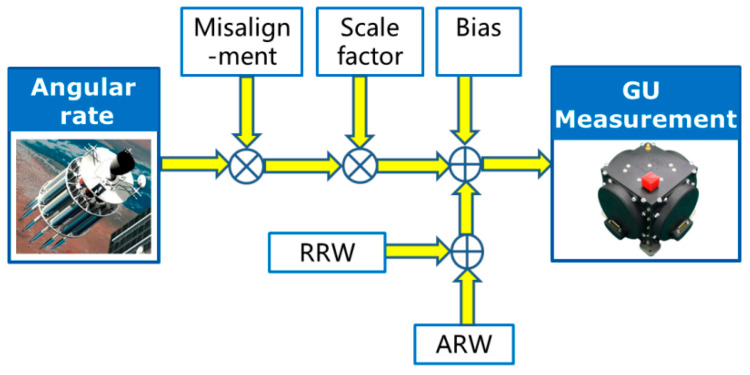
Diagram of gyroscope error model.

**Figure 3 sensors-25-02774-f003:**
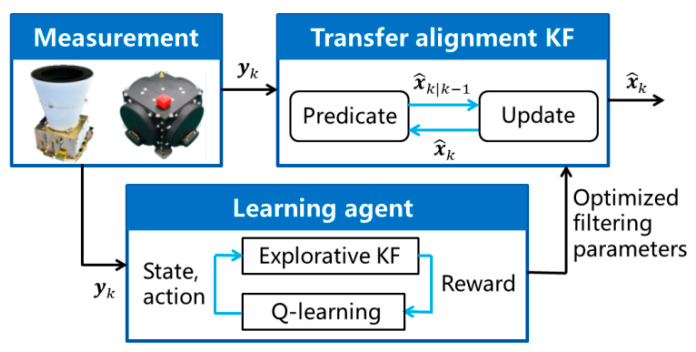
Q-learning Kalman filtering framework.

**Figure 4 sensors-25-02774-f004:**
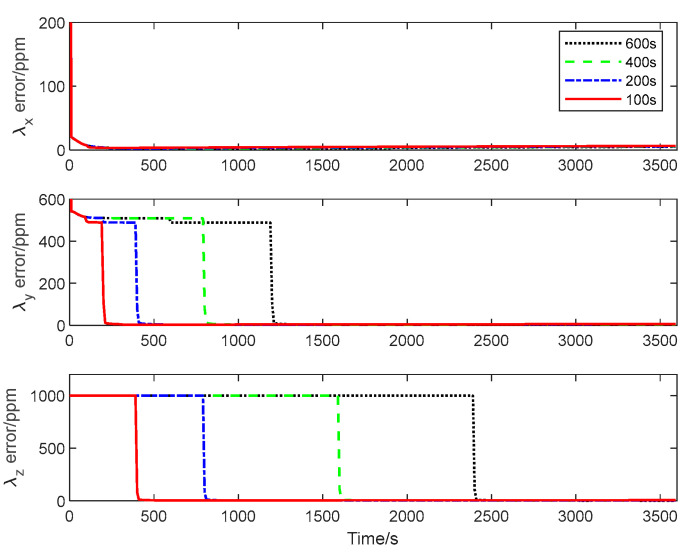
Estimation error bound for $\lambda_{xk}$, $\lambda_{yk}$ and $\lambda_{zk}$ with different maneuver time.

**Figure 5 sensors-25-02774-f005:**
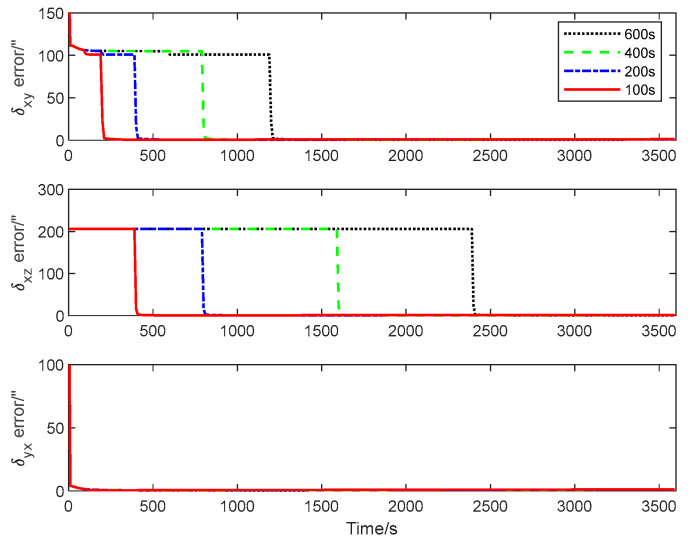
Estimation error bound for $\delta_{xyk}$, $\delta_{xzk}$ and $\delta_{yxk}$ with different maneuver time.

**Figure 6 sensors-25-02774-f006:**
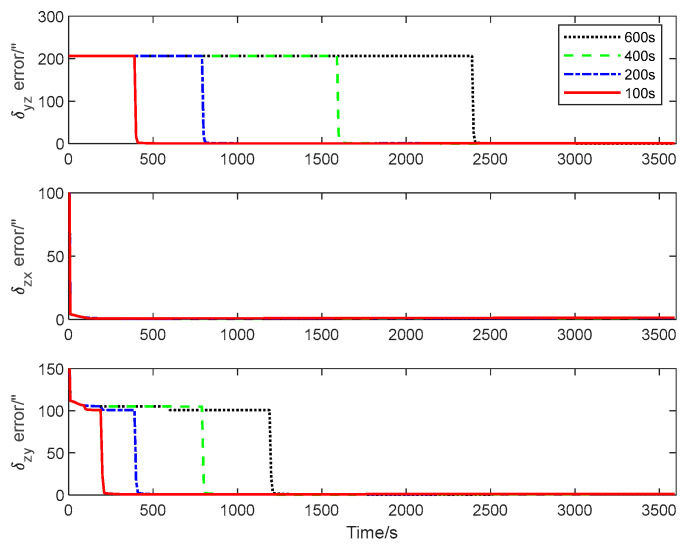
Estimation error bound for $\delta_{yzk}$, $\delta_{zxk}$ and $\delta_{zyk}$ with different maneuver time.

**Figure 7 sensors-25-02774-f007:**
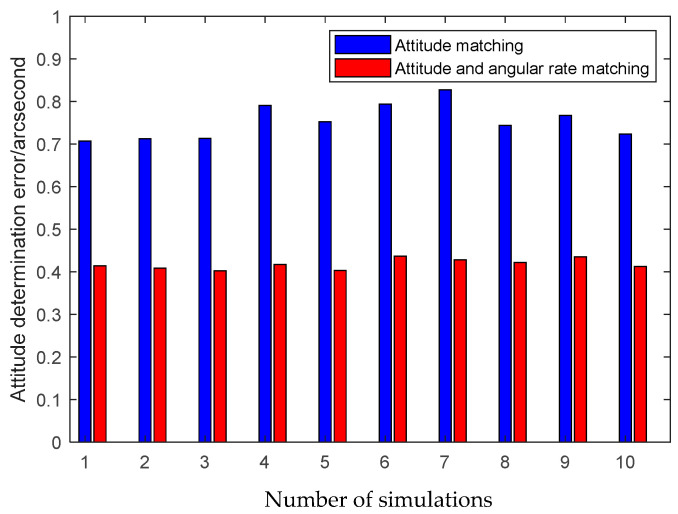
Attitude estimation errors of different measurement matching schemes.

**Figure 8 sensors-25-02774-f008:**
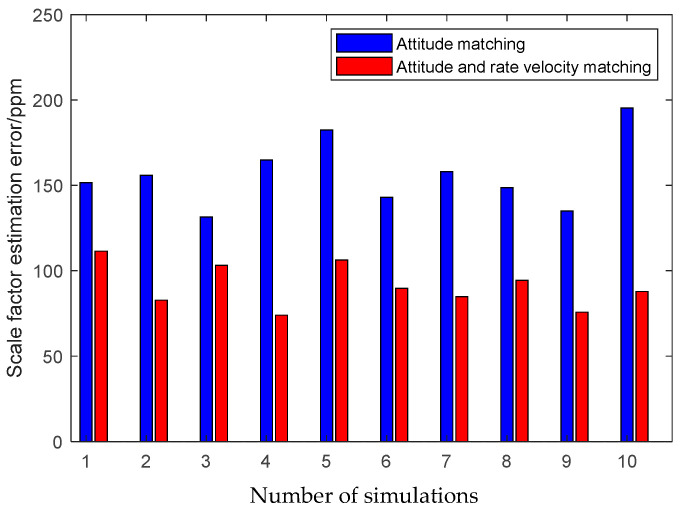
Scale factor estimation errors of different measurement matching schemes.

**Figure 9 sensors-25-02774-f009:**
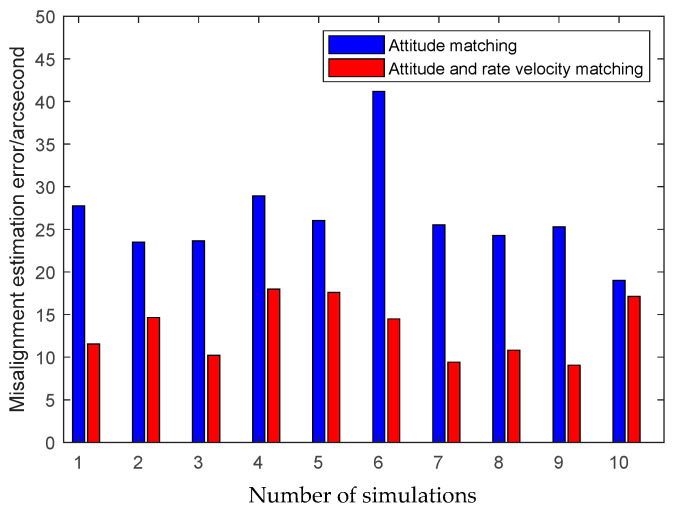
Misalignment estimation errors of different measurement matching schemes.

**Figure 10 sensors-25-02774-f010:**
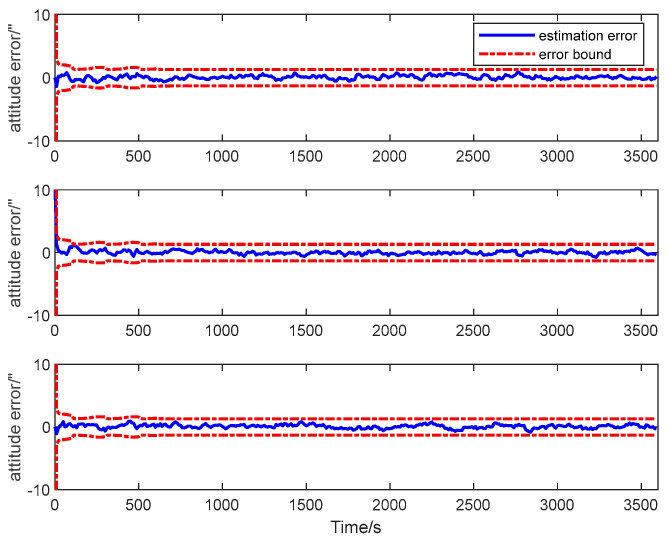
Attitude estimation error of transfer alignment based on QKF.

**Figure 11 sensors-25-02774-f011:**
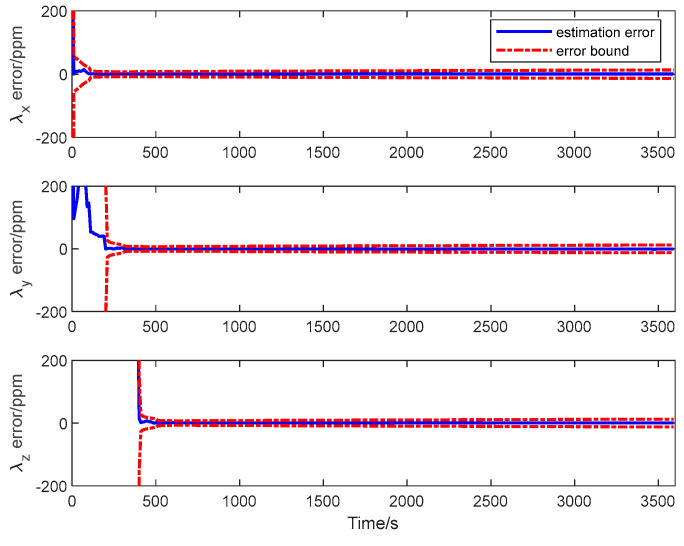
Scale factor estimation error of transfer alignment based on QKF.

**Figure 12 sensors-25-02774-f012:**
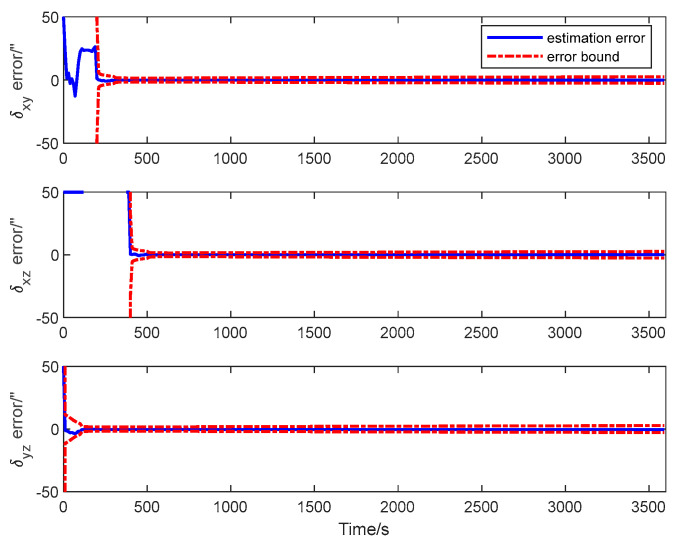
Misalignment estimation error of transfer alignment based on QKF.

**Figure 13 sensors-25-02774-f013:**
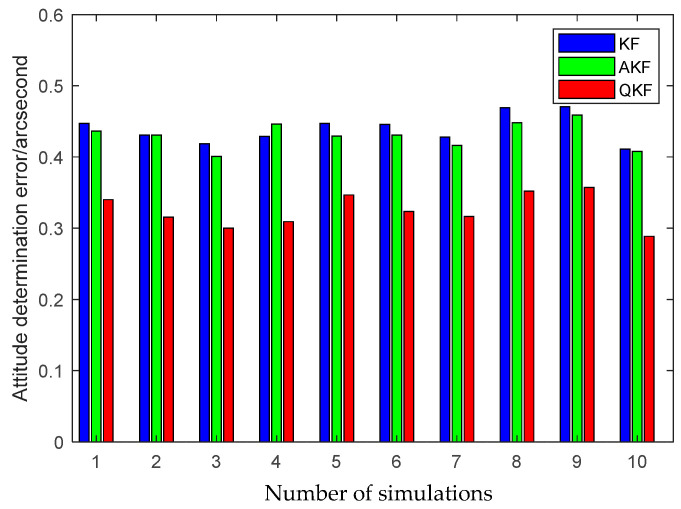
Attitude estimation errors of different filtering algorithms.

**Figure 14 sensors-25-02774-f014:**
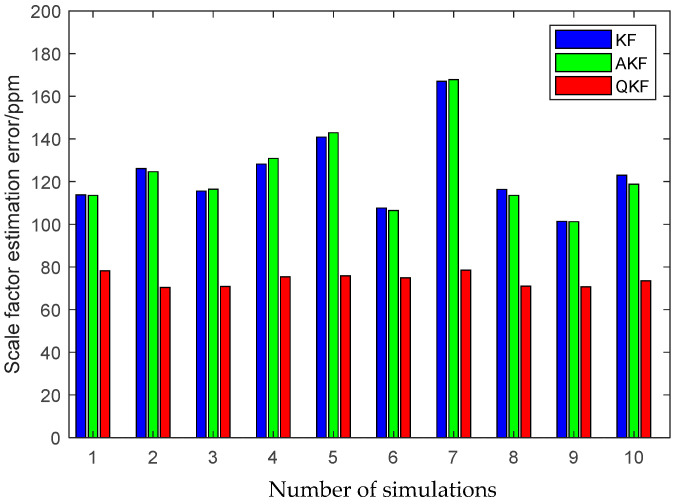
Scale factor estimation errors of different filtering algorithms.

**Figure 15 sensors-25-02774-f015:**
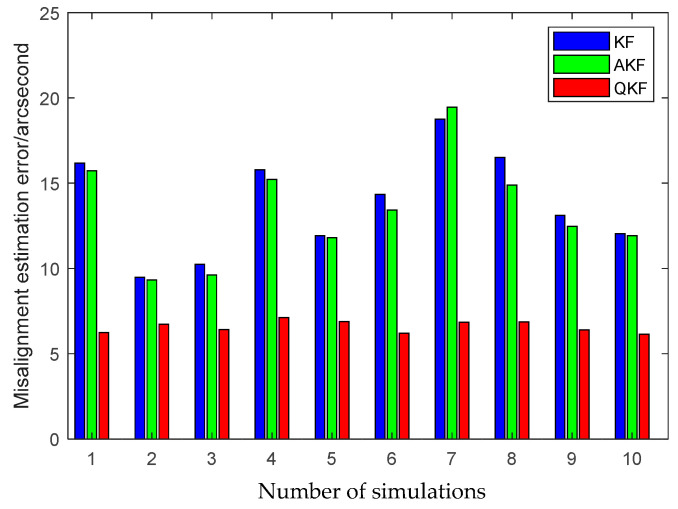
Misalignment estimation errors of different filtering algorithms.

**Figure 16 sensors-25-02774-f016:**
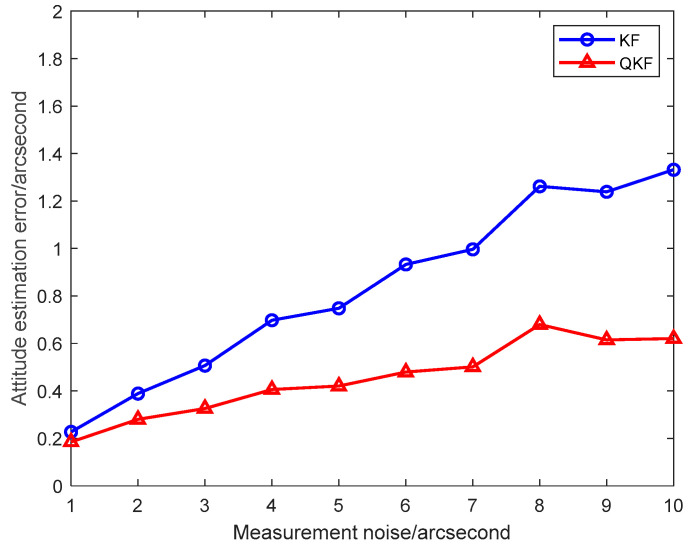
Attitude estimation errors of KF and QKF with different measurement noise.

**Table 1 sensors-25-02774-t001:** Average RMS error of state estimation error for transfer alignment.

Matching Scheme	Filtering Algorithm	Average RMS Error		
		Attitude (″)	Scale Factor (ppm)	Misalignment (″)
Attitude	KF	0.78	140.17	23.14
Attitude + angular rate	KF	0.42	113.40	13.91
Attitude + angular rate	QKF	0.31	69.80	6.30

## Data Availability

Data are contained within the article.

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
