# Peer review of "Inter-Spacecraft Rapid Transfer Alignment Based on Attitude Plus Angular Rate Matching Using Q-Learning Kalman Filter"

_sensors, 2025, doi:10.3390/s25092774_

Round 1
Reviewer 1 Report
Comments and Suggestions for Authors
- For Eq(2), (29), (34), add or pad zeros into non-diagonal part
- It is suggested to clearly mention the state inside state vector that author plan to estimate before line 196
- It is recommended to show the equation of each Q in Eq (29) based on the process noise wk in Eq (28).What kind of distribution is assumed for each eta in Eq (28)?
- For Eq (33), is \nu_rhok only contain the error of difference between master and slave? How about uncertainties of attitude information from master?
- Why alpha = 0.2, gamma = 0.8, epsilon = 0.5 being selected? Could author provide how should we select these value?
- What is ppm?
- Figure 7 to Figure 9 are not clear or confuse. What are being presented? Should it be errors that are being compared? But y-axis shows accuracy. Which is the proposed method? Same for Fig. 13 to Fig 15, should y-axis being as XXXX error?
- Same for Figure 7 to 9, Figure13 to 15. The error scale is degree, minute or arcsecond?
- For the GU on slave spacecraft, could author provide an example model? Since if objective is for Cubesat, the specification of gyro (or MEMS gyro) will be much lower.
- Line 435, author need explain or justify why such Q-learning parameters are set. Based on what reasoning
- 11 and 12, recommend to use different scale in y-axis so that the steady state error boundaries can be clearly seen
- The simulation seems to focus on one specific scenario or configuration. What if there is a change in IMU performance? Which is typically happening due to discontinuity of certain commercial hardware in the market. And how resilience of QKF with respect to different parameters? E.g. noise level?
Reviewer 2 Report
Comments and Suggestions for Authors
Overall the paper is well organized. I have some suggestions for clarification or reference from the authors
1. Figures 10 and 11 should add labels for lines of different colors. As well as explain why ±3σ bounds are used
2. Q-learning algorithms consume a lot of time and space for both the lookup and storage of the Q table when applying them, how much computational resources of hardware are needed in the Inter-spacecraft Rapid Transfer Alignment problem? How to ensure the efficiency of the algorithm?
3. What is the biggest difficulty of the Rapid Transfer Alignment problem, and how is it solved in principle by the application of QKF, and is it applicable in some other fields? And, the difference with the existing Q-learning based Kalman state estimation algorithm needs to be further elaborated
4. How to verify the accuracy of the simulation model? Is it possible to do hardware-in-the-loop testing?
5. The limitations of the study and the remaining challenges for real-time applications should be mentioned.
Reviewer 3 Report
Comments and Suggestions for Authors
The paper deals with the problem of transferring and calibrating parameters between the master and slave spacecraft before their separation. There are several comments on the article that could improve it.

Round 2
Reviewer 1 Report
Comments and Suggestions for Authors
no further comment